# Oat Polar Lipids Improve Cardiometabolic-Related Markers after Breakfast and a Subsequent Standardized Lunch: A Randomized Crossover Study in Healthy Young Adults

**DOI:** 10.3390/nu13030988

**Published:** 2021-03-18

**Authors:** Mohammad Mukul Hossain, Juscelino Tovar, Lieselotte Cloetens, Maria T. Soria Florido, Karin Petersson, Frederic Prothon, Anne Nilsson

**Affiliations:** 1Department of Food Technology, Engineering and Nutrition, Lund University, P.O. Box 124, 221 00 Lund, Sweden; juscelino.tovar@food.lth.se (J.T.); anne.nilsson@food.lth.se (A.N.); 2Division of Pure and Applied Biochemistry, Lund University, P.O. Box 124, 221 00 Lund, Sweden; lieselotte.cloetens@tbiokem.lth.se (L.C.); maria.soria_florido@tbiokem.lth.se (M.T.S.F.); 3Oatly AB, 261 51 Landskrona, Sweden; karin.petersson@oatly.com (K.P.); frederic.prothon@oatly.com (F.P.)

**Keywords:** appetite regulation, blood lipids, glycemic regulation, GLP-1, metabolic regulation, oat, polar lipids, postprandial glucose response, PYY, RCT

## Abstract

It has been suggested that intake of polar lipids may beneficially modulate various metabolic variables. The purpose of this study was to evaluate the effect of oat polar lipids on postprandial and second meal glycemic regulation, blood lipids, gastrointestinal hormones, and subjective appetite-related variables in healthy humans. In a randomized design, twenty healthy subjects ingested four liquid cereal-based test beverages (42 g of available carbohydrates) containing: i. 30 g of oat oil with a low concentration (4%) of polar lipids (PLL), ii. 30 g of oat oil containing a high concentration (40%) of polar lipids (PLH), iii. 30 g of rapeseed oil (RSO), and iv. no added lipids (NL). The products were served as breakfast meals followed by a standardized lunch. Test variables were measured at fasting and during 3 h after breakfast and two additional hours following a standardized lunch. PLH reduced glucose and insulin responses after breakfast (0–120 min) compared to RSO, and after lunch (210–330 min) compared to RSO and PLL (*p* < 0.05). Compared to RSO, PLH resulted in increased concentrations of the gut hormones GLP-1 and PYY after the standardized lunch (*p* < 0.05). The results suggest that oat polar lipids have potential nutraceutical properties by modulating acute and second meal postprandial metabolic responses.

## 1. Introduction

The pandemic of lifestyle-related diseases, such as obesity and type 2 diabetes (T2D), constitutes a major public health and economic challenge, since these illnesses have serious consequences, e.g., cardiovascular disease (CVD) and premature death [1]. About half a billion (9.3%) people worldwide have diabetes (2019), and the number is predicted to rise to 10.2% by 2030, whereof T2D is 90% of all cases [1]. Thus, preventive strategies are urgently needed. 

It has been shown that bioactive components of whole grains may be associated with lower body mass index (BMI) and reduced risk of CVD and T2D [2,3]. In this regard, oats are rich in health-promoting components, such as soluble dietary fiber (beta-glucan), lipids, and bioactive phenolic compounds (avenanthramides, tocopherols) [4,5,6,7]. Many studies have demonstrated that regular consumption of oat products rich in beta-glucans is linked to reduction of low-density lipoprotein (LDL) cholesterol levels in blood and postprandial glycemia, which has led to health claims on beta-glucans containing products [4,8,9,10,11]. Moreover, oats are particularly rich in polar lipids, which constitute approximately 15 wt% of total lipids. The oat polar lipids include phospholipids, glycolipids, and sphingolipids. The most abundant polar lipids in oat are the galactolipids, mainly digalactosyldiacylglycerol (DGDG) and its derivatives with different polarity (TriGDG, TetraGDG, and various types of estolides) [12,13,14,15]. These galactolipids in plants constitute the bulk of thylakoid membrane lipids, and provide a lipid bilayer matrix for photosynthetic complexes as the main constituents [12].

The gut hormones glucagon-like peptide 1 (GLP-1) and peptide YY (PYY) are released in response to food intake, and are important in metabolic and appetite regulation [16]. It has been reported that dietary lipids decrease the gastric emptying rate, and are a potent stimulus for the release of gut hormones, thus having the potential to lower postprandial glucose response and induce satiety [17]. There is a lack of studies investigating the effects of extracted oat polar lipids on metabolic variables. It has been suggested that polar lipid fractions may be involved in the cholesterol-lowering effects of oats [18]. Ohlsson et al. (2014) studied the effects on plasma lipids and gut hormones after intake of liposomes made from fractionated oat oils [19]. It was observed that a breakfast meal containing oat oil liposomes significantly increased plasma concentrations of certain gut hormones, prolonged elevation of triglycerides, decreased free fatty acids (FFA), and reduced voluntary energy intake during the rest of the day. However, there is only scarce information regarding the effects of oat polar lipids on cardio-metabolic risk-related markers in the acute and second meal postprandial period perspective.

The aim of this study was to investigate the metabolic effects of oat polar lipids consumed at breakfast on glycemic responses, appetite sensations, circulating gut hormones, and the lipid profile in the postprandial phases following two consecutive meals (test breakfast and standardized lunch). For this purpose, a randomized crossover meal study was performed in healthy young adults. 

## 2. Materials and Methods

### 2.1. Study Subjects

Recruitments of test subjects took place between January and March 2019, and the clinical phase took place between March and May 2019. The inclusion criteria were age between 20–40 years, BMI between 19–25 kg/m^2^, being a non-smoker, and having no diagnosed metabolic disorders or food allergies. Each subject got a full explanation (written and oral) of the purpose and protocol of the study, and written informed consent was obtained. Volunteers were aware of the possibility of withdrawing from the study at any time. Twenty healthy volunteers, five men and 15 women, aged (mean ± SEM) 24.8 ± 0.9 years, and with BMI (mean ± SEM) 22.2 ± 0.4 kg/m^2^ were enrolled to participate in the study.

### 2.2. Study Design and Procedure

The study was performed using a single-blind crossover design. Each participant consumed all of the meals, and in a random order. The different products were tested approximately one week apart. The day before each experiment, the participants had to avoid strenuous exercise, alcohol intake, and the consumption of oat products and food with high fiber content (beans, whole grain bread, fiber enriched pasta, whole cereal kernels, etc.). No antibiotics or probiotics were allowed within two weeks before and during the study period. To keep track of standardization of meal patterns prior to each experiment day, the participants were requested to provide a meal record from the day before each experiment day. Furthermore, the participants were instructed to consume individualized dinner meals at 18:00 on the day before each experiment. They consumed the same dinner on all occasions. In addition, the participants consumed a standardized evening meal at 21:00, consisting of a commercial white wheat bread with topping of their choice (same topping before each experimental day). After this meal, they fasted until a test breakfast meal was served at the research unit. Small quantities (half a glass) of water were, however, allowed prior to bedtime and when they woke up. The participants had to keep to the same routine before each experimental day. 

The subject arrived at the research unit at 07:30 after overnight fasting. Finger-prick blood samples and subjective appetite scores were obtained prior to the breakfast, which was served at 08:00 (time = 0 min, baseline). After that, the participants were instructed to consume the test meal within 12 min. For glucose measurements, additional capillary blood samples were drawn at 15, 30, 45, 60, 90, 120, 150, 180, and 210 min after the start of the breakfast meal. After the blood test, at 210 min, the standardized lunch was served, and blood samples were drawn at 225, 240, 255, 270, 300, and 330 min after baseline. Insulin was measured at the same time points as glucose, excluding 15 and 225 min. Considering the limited amount of capillary blood that can be drawn, triglycerides and gastrointestinal hormones were measured only at 0, 60, 210, and 330 min. During the experimental days, participants remained at the clinical facility, quietly seated, and were not allowed to eat or drink anything, except for the breakfast and lunch meals provided. 

### 2.3. Test Meals

The test meals were provided as liquid breakfasts, and consisted of an oat-based preparation specially designed for this study, added with different types and amounts of lipids (developed and manufactured by Oatly AB, Sweden). A glucose solution was included as the reference product. The oat base preparation was added with: i. oat oil containing low concentrations of polar lipids (PLL), ii. oat oil containing high concentrations of polar lipids (PLH), or iii. rapeseed oil (RSO) as a reference oil. Their metabolic effects were compared to those of the oat base preparation with no added lipids (NL). The oat oils were provided by Swedish Oat Fiber AB (Bua, Sweden), and rapeseed oil was purchased from AAK AB (Karlshamn, Sweden). All of the breakfast meals containing added lipids supplied an equivalent amount of total fat (33 g). All breakfast beverages contained 42 g of available carbohydrates. The nutritional composition of the breakfast meals is displayed in Table 1.

### 2.4. Standardized Lunch

The standardized lunch consisted of 122 g of white wheat bread (Pågen AB, Malmö, Sweden), corresponding to 42 g of available starch, 100 g of meatballs (Scan AB, Stockholm, Sweden), and 250 mL of water. Based on the nutritional facts declared by the producers, the total energy content of the lunch meal was 537 kcal. Table 2 shows the macronutrient composition of the lunch meal. 

### 2.5. Physiological Test Parameters

Plasma glucose concentrations were determined in whole blood using a HemoCue Glucose 201^+^ analyzer (HemoCue AB, Ängelholm, Sweden). Samples for serum insulin, serum free fatty acids (FFAs), and serum triglycerides (TGs) analyses were collected in BD Microtainer SST tubes. The tubes were left at room temperature for approximately 30 min and centrifuged for 5 min (5000 rpm, 25 °C, Eppendorf mini spin, F-45-12-11). The serum was then frozen at −40 °C until analysis. Additional blood plasma samples for GLP-1, GIP, PYY, and ghrelin analyses were collected in BD Microtainer K2E tubes. An inhibition cocktail consisting of DPP-4 inhibitor (10 µL/mL blood) (Millipore, St. Charles, MO, USA) and aprotinin (50 µL/mL blood) (Sigma-Aldrich, St. Louis, MO, USA) was added to the tube. The tubes were kept on ice before and after sampling, and then centrifuged for 10 min (4200 rpm, 4 °C) immediately after collecting the blood. Plasma samples were then frozen at −40 °C until analysis.

The determination of serum insulin concentrations was performed using a solid phase two-site enzyme immunoassay kit (Insulin ELISA 10-1113-01, Mercordia AB, Uppsala, Sweden). FFA concentrations were analyzed with an enzymatic colorimetric method with a 96 microplate (NEFA-HR (2) ACS-ACOD method, FUJIFILM Wako Chemicals Europe GmbH, Neuss, Germany). Serum triglyceride concentrations were determined by using a multi-sample enzymatic assay (LabAssay™ Triglyceride 290-63701, GPO.DAOS method, FUJIFILM Wako Chemicals Europe GmbH, Germany). The quantitative determination of total plasma GLP-1, PYY, GIP, and Ghrelin concentrations was performed by using a 10 spot U-plex assay kit (Meso Scale Diagnostics LLC, Rockville, MD, USA). 

A 100 mm visual analogue scale (VAS) was used to rank the subjective appetite variables satiety, hunger, and desire to eat. The left end of the VAS scale represented not at all (e.g., hungry), and right end of the scale represented extremely. The participants were told to rank the scale at every blood sampling point.

### 2.6. Statistical Analysis

Data are expressed as means ± SEM. A trapezoid model was used to calculate the incremental areas and areas under the curves (iAUCs and AUCs, respectively) for each subject and test meal. The iAUCs were used for statistical evaluations of blood glucose and insulin concentrations. Results presenting iAUCs of postprandial glucose and insulin responses after the standardized lunch (210–330 min) were calculated by considering the glucose concentrations prior to lunch (time = 210 min) as baseline. AUCs were used to present the results of appetite sensations, triglycerides, free fatty acids, ghrelin, GLP-1, PYY, and GIP. The plotting of graphs and calculations of areas was performed in GraphPad Prism (version 8.0, GraphPad Software, San Diego, CA, USA). 

Randomization of the consumption order of the test meals was performed by using random tools of Microsoft Excel (Washington, DC, USA). Differences in the test variables between the products (‘Meal’: NL, RSO, PLL, and PLH) at different time points during the experimental day (‘Time’) were evaluated using a mixed model (PROC MIXED in SAS release 9.4; SAS Institute Inc., Cary, NC, USA) with repeated measures and an autoregressive covariance structure for the test variables. Subjects were modelled as a random variable, and the corresponding baseline (fasting values) was modelled as the covariate. The effects of the test meals on physiological responses were evaluated using ANOVA (general linear model), followed by Tukey’s pairwise multiple comparison in MINITAB Statistical Software (version 18, Minitab, Minitab Inc., State College, PA, USA). Box Cox transformation was performed on the data prior to ANOVA analysis if the residuals were distributed unevenly (tested with Anderson–Darling, where *p* < 0.05 was considered unevenly distributed). 

If the value from a test subject was missing for one of the products, the test subject was excluded from the statistical evaluation of that specific test variable. Two subjects failed to follow the instructions at ranking appetite sensations, and these subjects were therefore excluded from the statistical analysis (*n* = 18). Due to missing samples, two subjects were excluded from GIP calculations (*n* = 18). The significance level was set at a *p*-value < 0.05.

## 3. Results

### 3.1. Postprandial Glucose and Insulin Responses after Consuming the Test Meals at Breakfast and Following the Standardized Lunch

#### 3.1.1. Evaluation of Acute Postprandial Glycemic Properties of the NL

To gain knowledge about the glycemic properties of the NL preparation, the postprandial (iAUC = 0–120 min) and second meal (iAUC = 210–330 min) glucose and insulin responses after the NL consumed at breakfast were compared with the responses after a glucose solution, containing similar amounts of available carbohydrates (42 g). No main effects of products on glucose (Figure 1) and insulin responses during the experimental session were detected between intake of the glucose solution and NL at breakfast. There were no differences at fasting prior to consumption of the glucose solution and the NL with respect to glucose concentrations (5.06 ± 0.07 and 4.91 ± 0.11 mmol/L, respectively, *p* > 0.05) or insulin concentrations (0.040 ± 0.004 and 0.0410 ± 0.004 nmol/L, respectively, *p* > 0.05). No significant differences between the glucose solution and the NL were observed regarding glucose responses (162 ± 13 and 149 ± 16 mmol*min/L, respectively, *p* > 0.05) and insulin responses (17.43 ± 1.84 and 18.10 ± 1.85 nmol/L, respectively, *p* > 0.05) in the postprandial period after intake at breakfast (iAUC = 0–120 min). Neither were there significant differences after the standardized lunch (iAUC = 210–330 min) between glucose and NL breakfasts with respect to glucose responses (210.3 ± 24.0 and 231.8 ± 19.1 mmol*min/L, *p* > 0.05) and insulin responses (21.24 ± 2.25 and 19.74 ± 2.32 nmol/L, respectively, *p* > 0.05) (Figure 1). 

Since no differences were observed between the glucose solution and NL with respect to postprandial glucose and insulin responses, only NL was included in further evaluations. 

#### 3.1.2. Evaluation of the Lipid-Supplemented Oat Preparations with Respect to Postprandial (Breakfast) and Second Meal (Lunch) Glucose and Insulin Responses

The results revealed significant main effects of meals on blood glucose responses and meal*time interactions (Figure 2). Lower postprandial blood glucose was observed after the test meal PLL and PLH compared to NL (*p* < 0.05) (iAUC = 0–120 min). In addition, the glucose iAUC (0–120 min) after the breakfast meal with PLH was lower compared to RSO (*p* < 0.01). Regarding the glycemic response to the standardized lunch (iAUC = 210–330 min), PLH breakfast resulted in significantly lower glycemic response compared with the breakfasts containing NL (*p* < 0.001), PLL (*p* < 0.01), and RSO (*p* < 0.001, Table 3).

The main effects of meal and meal*time were also observed on insulin responses during the experimental period (iAUC = 0–330 min; Figure 3). The breakfast postprandial insulin responses (iAUC = 0–120 min) were significantly lower after PLH compared to NL (*p* < 0.001) and RSO (*p* < 0.01). The insulin responses to the standardized lunch (iAUC = 210–330 min) were similar regardless of the test meal consumed at breakfast (*p* > 0.05, Table 3). No significant differences were observed concerning fasting glucose or fasting insulin concentrations before the start of the breakfast meals (*p* > 0.05, Table 3).

### 3.2. Triglycerides

Significant main effects of meals on TG and meal*time effects were found in the postprandial period after breakfast (0–210 min) (Figure 4). Intake of PLH resulted in lower TG concentrations compared to RSO and PLL. The AUCs (0–210 min) were significantly higher after RSO (*p* < 0.05) and PLL (*p* < 0.01) compared to after NL. At the end of the experimental session (330 min), the concentration of TG was significantly lower after the RSO breakfast compared with after the NL breakfast (*p* < 0.05, Table 4). No significant differences were observed between fasting TG concentrations before the start of the breakfast meals (*p* > 0.05, Table 4).

### 3.3. Free Fatty Acids

The results showed a significant main effect of meals as well as a meal*time effect during the experimental period (Figure 5). The FFA AUCs (0–210 min) were significantly lower after the PLH breakfast meal compared to RSO (*p* < 0.05) and PLL (*p* < 0.01). After the standardized lunch, reduced FFA AUCs (210–330 min) were observed after the PLH breakfast compared to PLL (*p* < 0.01) and RSO (*p* < 0.01). No significant differences in FFA concentrations were observed at fasting prior to the start of the breakfast meals (*p* > 0.05, Table 4).

### 3.4. Ghrelin

Figure 6 shows main effects of meals and meal*time effects after the test breakfast (0–210 min). The results revealed that the ghrelin concentrations were significantly lower after breakfasts (0–210 min) composed of PLH compared to NL (*p* < 0.001). After the standardized lunch (330 min), no significant differences (*p* > 0.05) were observed between the test meals. No significant differences were observed between fasting ghrelin concentrations before the start of the breakfast meals (*p* > 0.05, Table 5).

### 3.5. GLP-1

The results regarding GLP-1 are presented in Figure 7 and Table 5. Significant meals and meal*time effects were detected in the postprandial period after breakfast (0–210 min). The GLP-1 concentrations (AUC = 0–210 min) were significantly increased after the breakfast composed of PLH compared to NL (*p* < 0.001) and RSO (*p* < 0.001). In addition, PLH breakfast resulted in increased GLP-1 concentrations in the end of the experimental period, time = 330 min, compared to NL (*p* < 0.01), PLL (*p* < 0.05), and RSO (*p* < 0.01) breakfasts. No significant differences were observed between fasting GLP-1 concentrations before the start of the breakfast meals (*p* > 0.05).

### 3.6. PYY

Results regarding PYY concentrations are presented in Figure 8 and Table 5. The results showed significant main effects of meal and significant meal*time in the postprandial period after breakfast (0–210 min). The PYY concentrations after the PLH breakfast (AUC = 0–210 min) were significantly higher compared to after the NL (*p* < 0.001) and RSO breakfasts (*p* < 0.01). Additionally, at the end of the test period (330 min), the PLH breakfast resulted in significantly higher concentrations of PYY compared to NL (*p* < 0.001), PLL (*p* < 0.01), and RSO breakfasts (*p* < 0.001). No significant differences in the fasting PYY concentrations were observed before the breakfast meals were served.

### 3.7. GIP

Postprandial GIP responses after the breakfast and standardized lunch meals are presented in the Figure 9 and Table 5. Significant main effects of meal and meal*time were observed in the postprandial period after breakfast (0–210 min), revealing higher GIP concentrations after PLL compared to after PLH (*p* < 0.01) and NL (*p* < 0.001) breakfasts. There were no significant differences in the fasting GIP concentrations before the breakfast meals were served.

### 3.8. Subjective Appetite Ratings

Appetite ratings after the meals and standardized lunches are presented in Figure 10 and Appendix A. No significant differences in appetite ratings (desire to eat, hunger, and satiety) were observed at fasting prior to the start of the breakfast meals (*p* > 0.05), nor were significant differences detected in appetite variables depending on meals after breakfast or after the standardized lunch (*p* > 0.05).

## 4. Discussion

This study investigated the metabolic effects of oat polar lipids in an acute breakfast and a second-meal standardized lunch meal setting. The dietary intake of polar lipids varies widely depending on the choice of diet. It has been estimated that 1–10% of total dietary lipids are polar lipids including all types of plant and animal sources, e.g., whole grains, nuts, vegetable oil, dairy products, fish, and meat [21]. The effects of oat polar lipids rich in glycolipids were compared with those of a widely used edible oil, i.e., rapeseed oil [22], which contains low amounts of glycolipids (<1%) [20]. The results indicate that supplementation of breakfast with oat polar lipids, particularly rich in galactosylacylglycerols, has the potential to improve cardio-metabolic risk-associated variables and increase the release of gut hormones in a 5.5 h time perspective after ingestion. 

Ingestion of polar lipids of both animal and plant origins has previously been shown to exert beneficial effects, e.g., with respect to inflammation and gut health. Most of these studies have been conducted to investigate the effects of milk polar lipids, and only a few studies have investigated the health effects of plant polar lipids [19,23,24,25,26,27,28,29]. These studies were carried out using low doses of polar lipids (i.e., <3 g) [23,24,25,26,28], or not focusing directly on their effects on postprandial glucose or insulin responses [19]. Therefore, to our knowledge, the present work is the first to observe improved postprandial glycemic regulation following the ingestion of a meal rich in oat polar lipids. Our results revealed that the breakfast meal containing 12 g of polar lipids from oats (PLH) reduced both glucose and insulin responses compared to rapeseed oil (RSO) or to an essentially fat-free breakfast (NL). In addition, PLH decreased after-lunch postprandial TG and FFA compared to NL and to RSO, respectively, as well as increased gut hormones involved in metabolic and appetite regulation (GLP-1 and PYY) compared to RSO. The beneficial effects of PLH also included an improved postprandial glycemic regulation and reduced FFA concentration following a lunch meal consumed 3.5 h after breakfast. Increased secretion of GLP-1 and PYY was also observed until the end of the experimental session (5.5 h). This constitutes an important finding, since a diet rich in high glycemic impact foods is an established risk factor for the development of T2D [30]. Furthermore, high levels of TG are an established risk factor for CVD [31,32]. Thus, the results indicate that oat polar lipids have the potential to improve multiple risk markers encompassed by the metabolic syndrome, and therefore can be suggested to be protective against T2D and CVD. 

In this study, oat polar lipids triggered the release of PYY and the incretin hormone GLP-1 in the gut, both of which have important implications in glycemic and appetite regulation. These hormones may be involved in a variety of mechanisms, including reduction of the gastrointestinal motility and gastric emptying rate, and as signal molecules in the gut–brain axis, resulting in satiety sensations. The results regarding the effects of oat polar lipids on gut hormones are thus in line with the improved postprandial glucose increments also observed here, both after breakfast and following lunch. Although fat in general reduces the acute gastric emptying rate, this study shows that the PLH meal has a more potent reducing effect on postprandial glucose than the other lipid-rich meals. These results suggest that oat polar lipids effectively reduce the postprandial glycemia after breakfast and a second meal. 

The concentration of ghrelin was significantly lower after the PLH breakfast compared to NL, but no significant differences were recorded between the lipid-containing products. A possible reason for this apparent inconsistency could be the small number of participants in the study, which might have been insufficient to detect significant differences. Nevertheless, only the PLH breakfast significantly reduced ghrelin concentrations compared to NL, indicating a higher potential for polar lipids to reduce hunger sensations compared to the other high-lipid products included in this study.

One possible underlying mechanism behind the increased concentrations of gut hormones relates to slower hydrolysis or limited digestion of oat polar lipids compared to e.g., rapeseed triglycerides, resulting in a delayed absorption, and thus potentially increased stimulation of gut hormones release (GLP-1 and PYY) throughout the gastrointestinal tract. Such a hypothetical mechanism was already suggested by Ohlsson et al. [19], and is supported by observations showing delayed enzymatic hydrolysis of polar lipids from oat and other sources in in vitro models [33,34]. GIP was another gut hormone investigated. Besides its initially described incretin action, a number of additional effects have been reported for this hormone, the physiological importance of which is not yet clear. Consequently, while GLP-1 is known to suppress the postprandial glucagon response, GIP has been suggested to exert the opposite effect, i.e., to promote an enhanced response [35]. In addition GIP has also been shown to facilitate fat accumulation in adipocytes [35]. Interestingly, PLH promoted a lower GIP response than the RSO meal. 

This study demonstrated that a breakfast meal containing 12 g of oat polar lipids resulted in a significantly lowered concentration of FFAs at the time of the standardized lunch meal. Similar observations were reported by Ohlsson et al. [19], and are in agreement with a putative delayed hydrolysis of lipids and prolonged absorption time of the digestion products. It could be suggested that the reduced late postprandial concentrations of serum FFAs promoted by the polar lipids, as compared with the other test meals, contributed to the improved glycemic response seen after lunch. Accordingly, it has been demonstrated that elevated circulating concentrations of FFAs correlate with impaired insulin signaling [36] and reduced glucose tolerance [37]. It is also in accordance with studies showing that a meal supplemented with plant-derived diacylglycerol has the potential to improve postprandial lipid profiles (by reducing VLDL cholesterol) and insulin sensitivity [38]. Reduced circulating concentrations of FFAs and TGs are important effects of dietary polar lipids, since elevated concentrations are considered a cardiovascular risk factor [32,39,40]. However, the lower blood lipid concentration observed here after the intake of PLH might also relate to differences in the actual TG concentration in the different oils tested. All test meals (with the exception of the NL) contained 33 g of added lipids. However, supplementation with PLH provides a significant proportion, 40%, of polar lipids and approximately 60% TG, while the PLL contained only 4% polar lipids, and the remainder were mainly TGs; RSO is essentially made of TGs. Therefore, the total amount of fatty acids ingested, and thus potentially absorbable as substrates for endogenous TG formation, was lower in the PLH breakfast.

A limitation in this study was the relatively low number of subjects, which may have compromised the statistical power for some of the test variables. Another potential limitation was the unbalanced gender ratio, since 15 out of 20 participants were women. The combination of the breakfast products was formulated to evaluate the metabolic effects of oat polar lipids included in a drink. It would be interesting to investigate the effects of oat polar lipids added to more complex food matrices, e.g., a solid meal. The breakfast meals containing added lipids should be considered as high-fat meals, since 60% of their calorie content comes from dietary lipids. The reason for the high total fat content in the investigated meals was the limited availability of more concentrated oat polar lipid preparations. However, for future studies, we aim at including highly purified oat polar lipids to reach the amounts needed to achieve metabolic effects, without significantly increasing the total fat intake. 

In summary, this study shows the beneficial effects of polar lipids from oats on postprandial glycemic regulation, blood lipids, and gut hormones in healthy volunteers, which suggests antidiabetic and anti-obesogenic properties, with the potential to prevent cardio-metabolic diseases. Further studies are needed to elucidate the underlying mechanisms behind these promising health effects of oat lipids.

## Figures and Tables

**Figure 1 nutrients-13-00988-f001:**
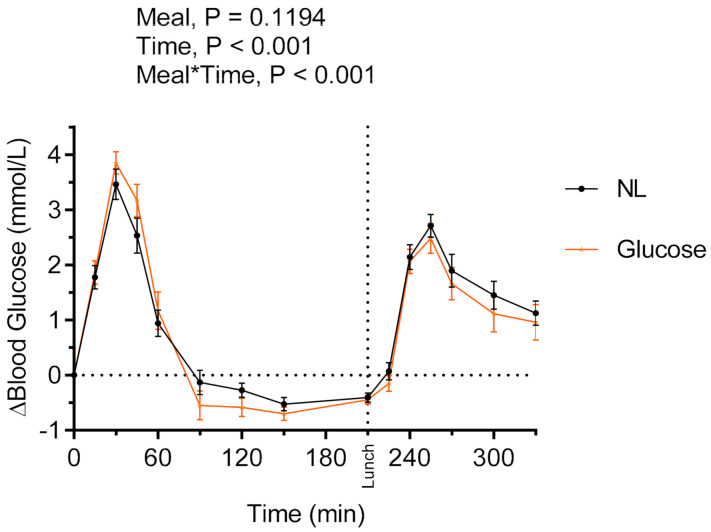
Incremental changes in serum blood glucose concentrations after test breakfasts and standardized lunch meals. Values are means ± SEM, *n* = 20 healthy subjects. Repeated measures; mixed model in SAS. NL, oat preparation without added oat polar lipids; glucose, glucose solution.

**Figure 2 nutrients-13-00988-f002:**
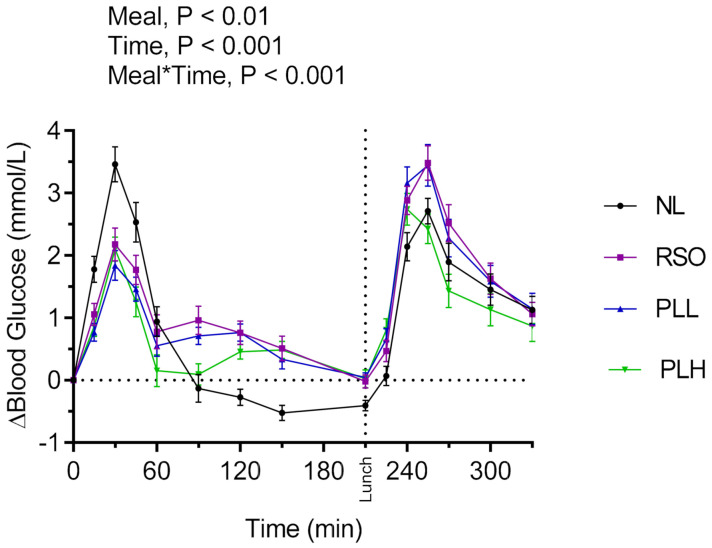
Incremental changes in serum blood glucose concentrations after test breakfasts and standardized lunch meals. Values are means ± SEM, *n* = 20 healthy subjects. Repeated measures; mixed model in SAS. NL, oat preparation without added lipids; RSO, oat preparation added with rapeseed oil; PLL, oat preparation with a low concentration of polar lipids; PLH, oat preparation with a high concentration of polar lipids.

**Figure 3 nutrients-13-00988-f003:**
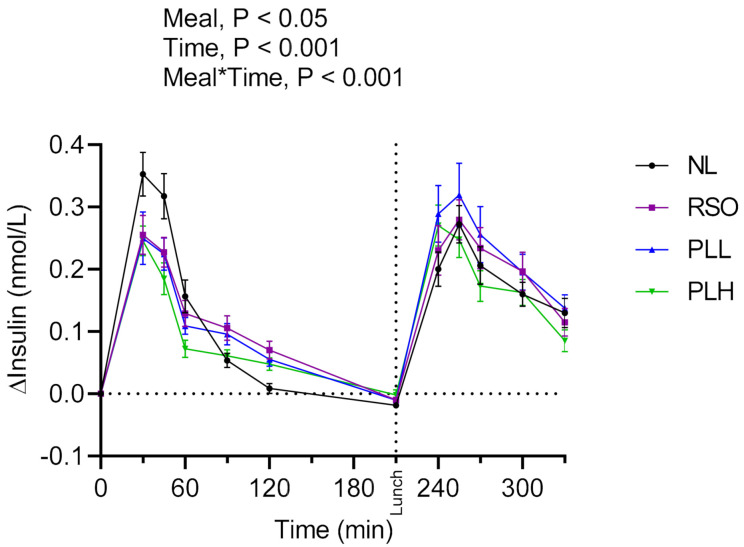
Postprandial incremental changes in blood serum insulin concentrations after test breakfasts and standardized lunch meals. Values are means ± SEM, *n* = 20 healthy subjects. Repeated measures; mixed model in SAS. NL, oat preparation without added lipids; RSO, oat preparation added with rapeseed oil; PLL, oat preparation with a low concentration of polar lipids; PLH, oat preparation with a high concentration of polar lipids.

**Figure 4 nutrients-13-00988-f004:**
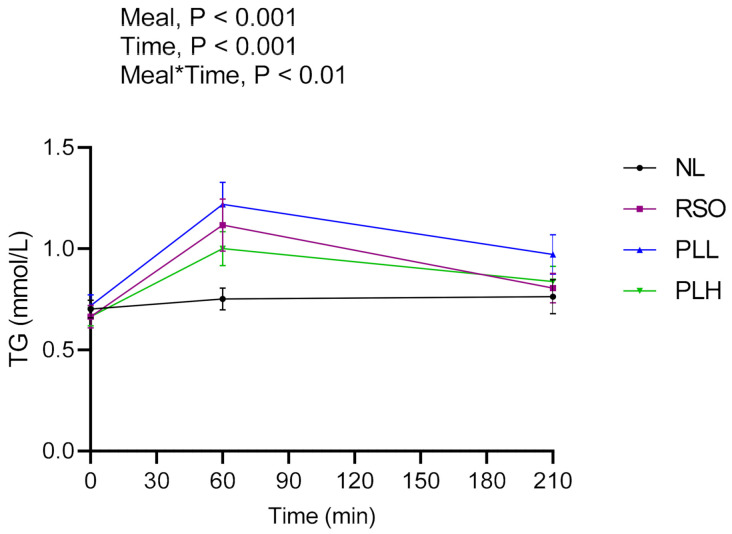
Concentration of serum triglycerides (TGs) after the breakfast meal. Values are means ± SEM, *n* = 20 healthy subjects. Repeated measures; mixed model in SAS. NL, oat preparation without added lipids; RSO, oat preparation added with rapeseed oil; PLL, oat preparation with a low concentration of polar lipids; PLH, oat preparation with a high concentration of polar lipids.

**Figure 5 nutrients-13-00988-f005:**
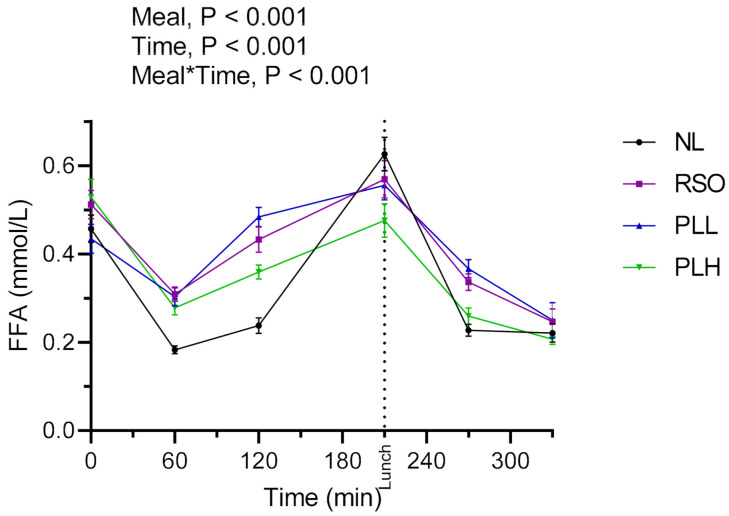
Concentrations of serum free fatty acid (FFAs) after breakfast and a standardized lunch meal. Values are means ± SEM, *n* = 20 healthy subjects. Repeated measures; mixed model in SAS. NL, oat preparation without added lipids; RSO, oat preparation added with rapeseed oil; PLL, oat preparation with a low concentration of polar lipids; PLH, oat preparation with a high concentration of polar lipids.

**Figure 6 nutrients-13-00988-f006:**
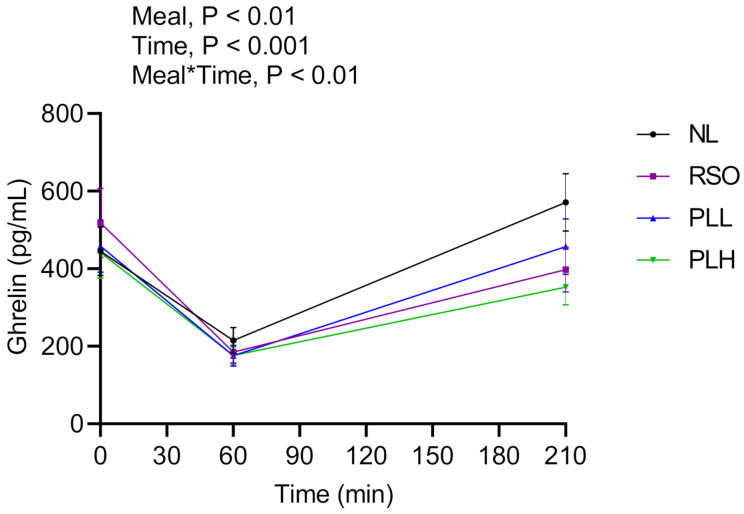
Mean concentrations of ghrelin after the breakfast meals. Values are means ± SEM, *n* = 20 healthy subjects. Repeated measures; mixed model in SAS. NL, oat preparation without added lipids; RSO, oat preparation added with rapeseed oil; PLL, oat preparation with a low concentration of polar lipids; PLH, oat preparation with a high concentration of polar lipids.

**Figure 7 nutrients-13-00988-f007:**
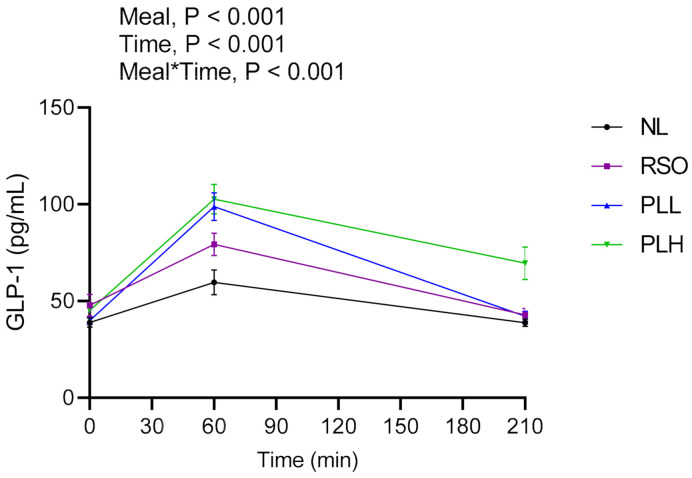
Mean concentrations of GLP-1 after the breakfast meal. Values are means ± SEM, *n* = 20 healthy subjects. Repeated measures; mixed model in SAS. NL, oat preparation without added lipids; RSO, oat preparation added with rapeseed oil; PLL, oat preparation with a low concentration of polar lipids; PLH, oat preparation with a high concentration of polar lipids.

**Figure 8 nutrients-13-00988-f008:**
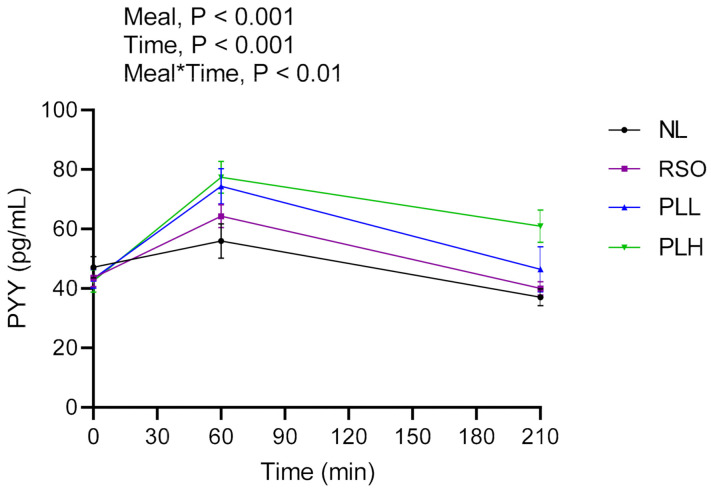
Mean concentration of PYY after the breakfast meal. Values are means ± SEM, *n* = 20 healthy subjects. Repeated measures; mixed model in SAS. NL, oat preparation without added lipids; RSO, oat preparation added with rapeseed oil; PLL, oat preparation with a low concentration of polar lipids; PLH, oat preparation with a high concentration of polar lipids.

**Figure 9 nutrients-13-00988-f009:**
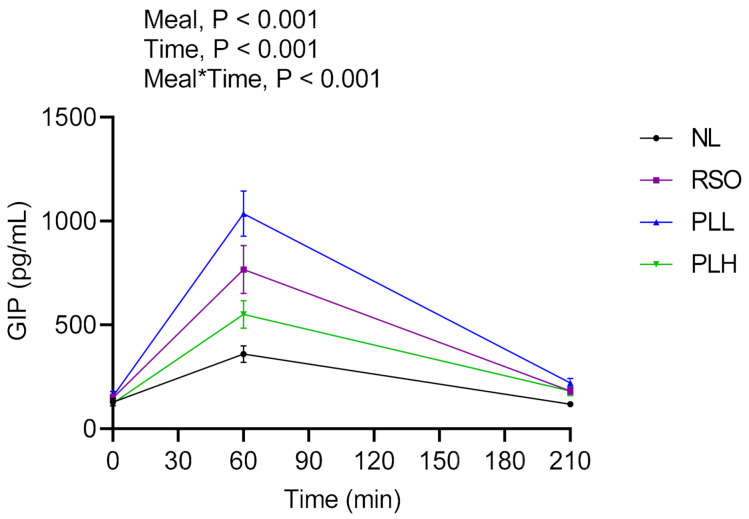
Mean concentrations of GIP after intake of test meals at breakfast. Values are means ± SEM, *n* = 18 healthy subjects. Repeated measures; mixed model in SAS. NL, oat preparation without added lipids; RSO, oat preparation added with rapeseed oil; PLL, oat preparation with a low concentration of polar lipids; PLH, oat preparation with a high concentration of polar lipids.

**Figure 10 nutrients-13-00988-f010:**
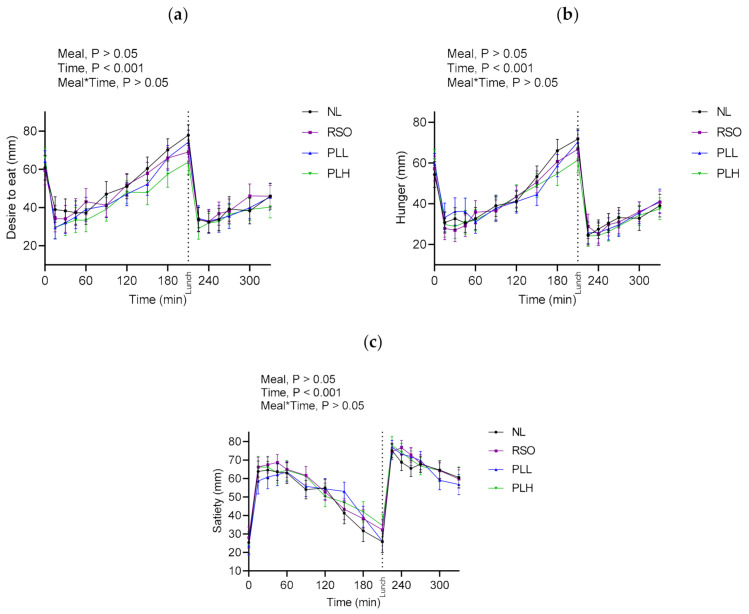
Subjective appetite ratings after test breakfasts and standardized lunch meals. Values are means ± SEM of subjective appetite ratings (VAS) of (**a**) desire to eat, (**b**) hunger, and (**c**) satiety; *n* = 20 healthy subjects. Repeated measures; mixed model in SAS. NL, oat preparation without added lipids; RSO, oat preparation added with rapeseed oil; PLL, oat preparation with a low concentration of polar lipids; PLH, oat preparation with a high concentration of polar lipids.

**Table 1 nutrients-13-00988-t001:** Nutritional composition of the breakfast meals per serving (500 mL) ^1^.

	PLL	PLH	RSO	NL	Glucose
Available Carbohydrates (g)	42	42	42	42	42
Free glucose (g)	1.75	1.75	1.75	1.75	42
Fat (g)	33	33	33	3	0
Polar lipids (g) ^2^	1	12	1	<0.5	0
Protein (g)	6.5	6.5	6.5	6.5	0
Dietary fiber (g)	5	5	5	5	0
beta-glucans (g)	2.5	2.5	2.5	2.5	0
Energy (Kcal)	491	491	491	221	168

^1^ PLL, oat preparation with a low concentration of oat polar lipids; PLH, oat preparation with a high concentration of oat polar lipids; RSO, oat preparation with added rapeseed oil; NL, oat preparation without added lipids. ^2^ Polar lipid contents in PLL and PLH according to the supplier, and in NL and RSO estimated according to reference [20].

**Table 2 nutrients-13-00988-t002:** Nutritional composition of lunch meal per 100 g ^1^.

	Meatballs	Bread
Carbohydrate (g)	8	47.0
Fat (g)	15	3.5
Protein (g)	13	8.5

^1^ Based on the nutritional facts declared by the producers.

**Table 3 nutrients-13-00988-t003:** Blood glucose and insulin concentrations at fasting and after consumption of test meals at breakfast, followed by a standardized lunch meal ^1^.

Test Variables	NL	RSO		PLL		PLH	
			%∆ ^2^		%∆ ^2^		%∆ ^2^
Glucose							
Fasting blood glucose (mmol/L)	4.91 ± 0.11 ^a^	4.85 ± 0.10 ^a^	−1.22	4.96 ± 0.09 ^a^	1.01	4.99 ± 0.09 ^a^	1.62
Blood glucose prior to std. lunch (at 210 min) (mmol/L)	4.4 ± 0.11 ^a^	4.8 ± 0.12 ^a^	8.07	4.9 ± 0.08 ^a^	11.65	5.0 ± 0.12 ^a^	12.78
Blood glucose iAUC = 0–120 min (mmol*min/L)	149 ± 16.3 ^a^	136 ± 16.4 ^ab^	−8.45	106 ± 11 ^bc^	−28.30	85 ± 12.9 ^c^	−42.58
Blood glucose iAUC = 210–330 min (mmol*min/L)	231.8 ± 19.1 ^a^	232.1 ± 23.3 ^a^	0.12	223.3 ± 21.1 ^a^	−3.66	166.4 ± 19.6 ^b^	−28.21
Blood glucose iAUC = 0–330 min (mmol*min/L)	335.5 ± 34.6 ^ab^	413.5 ± 39.5 ^a^	23.24	371.3 ± 36.9 ^ab^	10.67	293.9 ± 42.9 ^b^	−12.39
Insulin							
Fasting blood insulin (nmol/L)	0.041 ± 0.004 ^a^	0.039 ± 0.004 ^a^	−4.87	0.04 ± 0.003 ^a^	−2.43	0.04 ± 0.003 ^a^	−2.43
Blood insulin prior to std. lunch (at 210 min) (nmol/L)	0.023 ± 0.002 ^a^	0.030 ± 0.002 ^a^	30.43	0.03 ± 0.002 ^a^	30.43	0.041 ± 0.007 ^a^	78.26
Insulin iAUC = 0–120 (nmol*min/L)	18.1 ± 1.85 ^a^	16.3 ± 1.8 ^ab^	−9.94	15.16 ± 1.73 ^bc^	−16.24	12.51 ± 1.24 ^c^	−30.88
Insulin iAUC = 210–330 (nmol*min/L)	19.74 ± 2.32 ^a^	22.16 ± 2.97 ^a^	12.66	24.83 ± 3.51 ^a^	25.78	19.86 ± 2.22 ^a^	0.60
Insulin iAUC = 0–330 (nmol*min/L)	38.44 ± 4.04 ^ab^	41.59 ± 4.72 ^ab^	8.19	42.39 ± 5.14 ^a^	10.27	34.95 ± 3.56 ^b^	−9.07

^1^ Data are presented as means ± SEM, *n* = 20 healthy subjects. Values in the same row with different superscript letters are significantly different, *p* < 0.05 (ANOVA, followed by Tukey’s test). ^2^ The percentage change is calculated as the difference from the NL. NL, oat preparation without added lipids; RSO, oat preparation added with rapeseed oil; PLL, oat preparation with a low concentration of polar lipids; PLH, oat preparation with a high concentration of polar lipids; iAUC, incremental area under the curve.

**Table 4 nutrients-13-00988-t004:** TG and FFA responses after breakfast (0–210 min) and a subsequent standardized lunch (210–330 min) ^1^.

Test variables	NL	RSO		PLL		PLH	
			%∆ ^2^		%∆ ^2^		%∆ ^2^
Triglycerides (TG)							
Fasting TG (mmol/L)	0.70 ± 0.04 ^a^	0.66 ± 0.05 ^a^	−5.71	0.71 ± 0.05 ^a^	1.428	0.66 ± 0.04 ^a^	−5.714
TG = AUC 0–210 min (mmol*min/L)	157.2 ± 10.5 ^a^	203.2 ± 20.6 ^b^	29.26	221.8 ± 19.5 ^b^	41.09	187.4 ± 13.9 ^ab^	19.21
TG at 330 min (mmol/L)	0.89 ± 0.05 ^a^	0.70 ± 0.05 ^b^	−20.64	0.77 ± 0.07 ^ab^	−12.93	0.75 ± 0.05 ^ab^	−15.38
Free Fatty Acid							
Fasting FFA (mmol/L)	0.45 ± 0.03 ^a^	0.51 ± 0.03 ^a^	13.33	0.43 ± 0.03 ^a^	−4.44	0.52 ± 0.04 ^a^	15.55
FFA = AUC 0–210 min (mmol*min/L)	70.7 ± 2.83 ^a^	91.94 ± 3.34 ^b^	30.04	92.51 ± 3.93 ^b^	30.84	80.92 ± 2.9 ^c^	14.45
FFA = AUC 210–330 min (mmol*min/L)	39.05 ± 1.86 ^ab^	44.65 ± 2.19 ^a^	14.34	46.15 ± 2.72 ^a^	18.18	36.04 ± 1.85 ^b^	−7.70

^1^ Data are presented as means ± SEM, *n* = 20 healthy subjects. Values in the same row with different superscript letters are significantly different, *p* < 0.05 (ANOVA, followed by Tukey’s test). ^2^ The percentage change is calculated as the difference from the NL. NL, oat preparation without added lipids; RSO, oat preparation added with rapeseed oil; PLL, oat preparation added with oat oil containing a low concentration of polar lipids; PLH, oat preparation added with oat oil containing a high concentration of polar lipids; AUC, area under the curve.

**Table 5 nutrients-13-00988-t005:** Plasma GLP-1, PYY, Ghrelin, and GIP response after breakfast and lunch ^1^.

Test Variables	NL	RSO		PLL		PLH	
			%∆ ^2^		%∆ ^2^		%∆ ^2^
Fasting plasma Ghrelin (pg/mL) ^3^	444.8 ± 63.1 ^a^	518.4 ± 89 ^a^	16.5^4^	458.4 ± 67.2 ^a^	3.05	441.6 ± 65.8 ^a^	−0.719
Ghrelin AUC = 0–210 (pg*min/mL) ^3^	78,698 ± 10,536 ^a^	64,764 ± 9599 ^b^	−17.70	66,503 ± 9950 ^ab^	−15.49	58,184 ± 7246 ^b^	−26.06
Plasma Ghrelin at 330 min (pg/mL) ^3^	255.8 ± 41 ^a^	251.7 ± 38 ^a^	−1.60	226.5 ± 36 ^a^	−11.45	232.4 ± 22 ^a^	−9.1^4^
Fasting plasma GLP-1 (pg/mL) ^3^	38.95 ± 2.37 ^a^	47.86 ± 5.63 ^a^	22.87	39.99 ± 2.12 ^a^	2.67	45.18 ± 4.06 ^a^	15.99
GLP-1 AUC = 0–210 (pg*min/mL) ^3^	10,345 ± 726 ^a^	12,973 ± 668 ^ab^	25.40	14,731 ± 825 ^bc^	42.397	17,357 ± 1050 ^c^	67.78
Plasma GLP-1 at 330 min (pg/mL) ^3^	52.9 ± 4.14 ^a^	56.71 ± 4.75 ^a^	7.20	59.93 ± 6 ^a^	13.28	79.83 ± 6.55 ^b^	50.90
Fasting plasma PYY (pg/mL) ^3^	47.1 ± 3.57 ^a^	43.65 ± 3.04 ^a^	−7.32	43.3 ± 3.21 ^a^	−8.06	42.55 ± 3.72 ^a^	−9.66
PYY AUC = 0–210 (pg*min/mL) ^3^	10,066 ± 802 ^a^	11,068 ± 597 ^ab^	9.95	12,594 ± 1187 ^bc^	25.11	13,972 ± 893 ^c^	38.80
Plasma PYY at 330 min (pg/mL) ^3^	43.6 ± 2.08 ^a^	44.48 ± 3.52 ^a^	2.01	45.06 ± 3.12 ^a^	3.3^4^	58.33 ± 4.09 ^b^	33.78
Fasting plasma GIP (pg/mL) ^4^	134.1 ± 18 ^a^	154 ± 17.4 ^a^	14.8^3^	160.8 ± 25.3 ^a^	19.91	119.4 ± 11.5 ^a^	−10.96
GIP AUC = 0–210 (pg*min/mL) ^4^	51,723 ± 4651 ^a^	98,274 ± 13,425 ^bc^	90.00	133,403 ± 13,243 ^c^	157.91	77,645 ± 8641 ^ab^	50.11
Plasma GIP at 330 min (pg/mL) ^4^	354.2 ± 125 ^a^	338.8 ± 139 ^a^	−4.3^4^	349.7 ± 178 ^a^	−1.27	363.8 ± 175 ^a^	2.71

^1^ Data are presented as means ± SEM, ^3^
*n* = 20, ^4^
*n* = 18 healthy subjects. Values in the same row with different superscript letters are significantly different, *p* < 0.05 (ANOVA, followed by Tukey’s test). ^2^ The percentage change is calculated as the difference from the NL. NL, oat preparation without added lipids; RSO, oat preparation added with rapeseed oil; PLL, oat preparation with a low concentration of polar lipids; PLH, oat preparation with a high concentration of polar lipids; AUC, area under the curve.

## Data Availability

The datasets analyzed during this study are available from the corresponding author on reasonable request.

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
