# Peer review of "Oat Polar Lipids Improve Cardiometabolic-Related Markers after Breakfast and a Subsequent Standardized Lunch: A Randomized Crossover Study in Healthy Young Adults"

_nutrients, 2021, doi:10.3390/nu13030988_

Round 1

Reviewer 1 Report

The whole paper is well written. The introduction provides a good, generalized background. The objective of the study is clearly defined. The results are easy to follow. Conclusions are supported by the results. My specific comments are listed below:

  • some words had a yellow background in my version of the manuscript, please checked
  • how exactly the crossover part of the study was designed? did each participant test each of the meals? Was the meal assigned randomly each time? Please specify
  • why rapeseed oil was selected as a reference oil? what is the composition of polar lipids in rapeseed in comparison to PLL or PLH? Some papers are describing polar lipids in Rapeseed but you stated in table 1 that in yours RSO was 0 g of PL. Please explain
  • Please consider adding the information about standardized dinner meals and standardized evening meal
  • Table 3 – what is the meaning of ‘a’, ‘b’, ‘c’ superscript letters in the Table?
  • Why some physiological parameters were measured till 210min and some till 300min?
  • Line 341: did you mean figure 6 here not figure 5?

Author Response

Dear Reviewer,

Thank you for giving us the opportunity to submit a revised draft of our manuscript titled “Oat Polar Lipids Improve Cardiometabolic-related Markers After Breakfast and a Subsequent Standardized Lunch; A Randomized Crossover Study in Healthy Young Adults” to Nutrients (Manuscript ID: nutrients-1110462). We appreciate the time and effort that you have dedicated to providing your valuable feedback on our manuscript. We are grateful to you for your insightful comments on our paper. We have been able to incorporate changes to reflect most of the suggestions provided by you. We have highlighted the changes within the manuscript.

Here is a point-by-point response to your comments and concerns.

Overall Comments: The whole paper is well written. The introduction provides a good, generalized background. The objective of the study is clearly defined. The results are easy to follow. Conclusions are supported by the results.

Response: We would like to thank you for your kind comments.

Minor specific comments:

Comment 1: some words had a yellow background in my version of the manuscript, please checked

Response: Thank you for pointing this out. The original submitted version did not contain yellow background. We believe that the editor’s office made some format change, and we guess they suggested us to put space between the letters. All this was revised.

Comment 2: how exactly the crossover part of the study was designed? did each participant test each of the meals? Was the meal assigned randomly each time? Please specify

Response: We have, accordingly, modified the description of the crossover design to emphasize this point and incorporated new text: “The study was performed using a single-blind crossover design. Each participant consumed all the meals, and in a random order”.  (P. 3 line 100-102)

Comment 3: why rapeseed oil was selected as a reference oil? what is the composition of polar lipids in rapeseed in comparison to PLL or PLH? Some papers are describing polar lipids in Rapeseed but you stated in table 1 that in yours RSO was 0 g of PL. Please explain

Response: You have raised an important point here. We agree with this comment. We therefore revised the polar lipid contents in Table 1 and added explanation in the discussion for the reasons for choosing rapeseed oil as a reference oil (P. 4 line 151-153 table 1; P. 17 line 447-449).

Comment 4: Please consider adding the information about standardized dinner meals and standardized evening meal

Response: Thank you for the suggestion. We have incorporated more information to clarify the standardized dinner and evening meals. (P. 3 line 110 – 116)

Comment 5: Table 3 – what is the meaning of ‘a’, ‘b’, ‘c’ superscript letters in the Table?

Response: The meaning of ‘a’, ‘b’, ‘c’ superscript were defined in the original manuscript and are found in the figure legends as “Values in the same row with different superscript letters are significantly different, P < 0.05”. (e.g. p.9)

Comment 6: Why some physiological parameters were measured till 210min and some till 300min?

Response: Thank you for pointing this out. A limitation in the study was the small volume of blood samples obtained, due to capillary blood sampling instead of venous blood sampling. Consequently, it was not possible to investigate all the variables at each sampling point. However, we have added additional information to specify each sampling point for individual physiological parameters. (p. 3 line 122-130)

Comment 7: Line 341: did you mean figure 6 here not figure 5?

Response: We apologized for this error. Revised as suggested. (p. 12 line 349)

We look forward to hearing from you and to respond to any further questions and comments you may have.

Sincerely,

Mohammad Mukul Hossain

Lund University, Sweden.

Reviewer 2 Report

This study examined the effects of consumption of oat polar lipids on several post-prandial metrics in a randomized crossover trial. They found that consumption of high density polar lipid oat product resulted in better post-prandial glycemia, insulinemia, lipidemia and gut hormones compared to the other test meals. Overall, this study was a well done evaluation on the direct effects of this nutrient on post-prandial metabolic profiles.

I have two suggestions for the methods and discussion. In section 2.2: More explanation on the cross-over aspect of this trial would be useful. At first read, I was unclear if each participant took part in every test meal. Did the trial take place over four weeks with each experiment at the beginning of each week? Also it seems there was a ~6 day washout period between each test meal. Can you clarify this information as well? It seems they were randomized as to the order of the meals (based on information in the statistical analysis section). It might be useful to include this information here as well.

In the discussion, my suggestion would be to add some public health context in the discussion. It would be helpful to provide an estimate of how much oat polar lipid and dietary polar lipid is generally being consumed now in the diet. What other foods provide polar lipids? It would be useful to include a few sentences as to the public health implications.

Minor edits:

-line 52: LDL needs to be spelled out at first usage

-line 63-69: Could use citations

 -Lines 89-98: Was this study approved by IRB? Would be useful to include here.

-Line 100: ‘e’ is missing from ‘The’

-Line 112: Confirm that they were instructed that they could only consume water after ‘X’ time the day before experiment

-line 208: ‘meals’ misspelled

Author Response

Dear Reviewer,

Thank you for giving us the opportunity to submit a revised draft of our manuscript titled “Oat Polar Lipids Improve Cardiometabolic-related Markers After Breakfast and a Subsequent Standardized Lunch; A Randomized Crossover Study in Healthy Young Adults” to Nutrients (Manuscript ID: nutrients-1110462). We appreciate the time and effort that you have dedicated to providing your valuable feedback on our manuscript. We are grateful to you for your insightful comments on our paper. We have been able to incorporate changes to reflect most of the suggestions provided by you. We have highlighted the changes within the manuscript.

Here is a point-by-point response to the reviewers’ comments and concerns.

Comments from Reviewer

Overall Comments: This study examined the effects of consumption of oat polar lipids on several post-prandial metrics in a randomized crossover trial. They found that consumption of high density polar lipid oat product resulted in better post-prandial glycemia, insulinemia, lipidemia and gut hormones compared to the other test meals. Overall, this study was a well done evaluation on the direct effects of this nutrient on post-prandial metabolic profiles.

Response: We would like to thank reviewer for this kind comments.

Minor specific comments:

Comment 1: In section 2.2: More explanation on the cross-over aspect of this trial would be useful. At first read, I was unclear if each participant took part in every test meal. Did the trial take place over four weeks with each experiment at the beginning of each week? Also it seems there was a ~6 day washout period between each test meal. Can you clarify this information as well? It seems they were randomized as to the order of the meals (based on information in the statistical analysis section). It might be useful to include this information here as well.

Response: Thank you for pointing this out. We have, accordingly, modified the crossover design to emphasize this point and incorporate new sentence “The study was performed using a single-blind crossover design. Each participant consumed all the meals, and in a random order”.  (P. 3 line 100-102).

Comment 2: In the discussion, my suggestion would be to add some public health context in the discussion. It would be helpful to provide an estimate of how much oat polar lipid and dietary polar lipid is generally being consumed now in the diet. What other foods provide polar lipids? It would be useful to include a few sentences as to the public health implications.

Response: We agree with your suggestion and therefore we have included few words regarding rough estimation and source of dietary polar lipids. (p. 17 line 449-453).

Comment 3: line 52: LDL needs to be spelled out at first usage.

Response: Revised as suggested. (P. 2 line 52).

Comment 4: line 63-69: Could use citations.

Response: Thank you for the suggestion and have incorporated relevant references. (P. 2 line 65, 68).

Comment 5: Lines 89-98: Was this study approved by IRB? Would be useful to include here.

Response: Thank you for your suggestion. We checked the possibility to get IRB approval. Unfortunately, the study requires prior approval. We will consider this for the upcoming studies. Just to let you know that the current study was registered at ClinicalTrials.gov (NCT03830736). 

Comment 6: Line 100: ‘e’ is missing from ‘The’

Response: Thanks. Done (p. 3, line 100).

Comment 7: Line 112: Confirm that they were instructed that they could only consume water after ‘X’ time the day before experiment.

Response: Revised as suggested. We have incorporated new sentence to emphasize this. (P. 3 line 116-120).

Comment 8: line 208: ‘meals’ misspelled.

Response: Thanks. Corrected. (p. 5 line 216).

We look forward to hearing from you in due time and to respond to any further questions and comments you may have.

Sincerely,

Mohammad Mukul Hossain

Lund University, Sweden.

Reviewer 3 Report

In this age, where one of the biggest challenge and threat at the same time to humans is the ill-effects of unhealthy eating habits. Due to this, ultimately the life expectancy of people is on a constant decrease. Impetus is provided on eating healthy food products, superfoods, etc. As such this study becomes very crucial and it is my belief that it will aid us in recognizing healthy substitutes contributing to the our overall well being. The study shows beneficial effects of polar lipids from oat on postprandial glycemic regulation, blood lipids and gut hormones in healthy volunteers. This study involved evaluation of the effect of oat polar lipids on postprandial and second meal glycemic regulation, blood lipids gastrointestinal hormones, and subjective appetite-related variables in healthy humans. In the process, various parameters such as serum blood glucose concentrations, blood serum insulin concentrations, serum triglycerides, serum free fatty acid, ghrelin, GLP-1, etc were evaluated after breakfast and lunch. Authors strict control over various experimental parameters during the study design is commendable and appreciated. I read this prospective manuscript with great interest and accept it after addressing minor revisions as follows:

  1. Introduction line 42: End of sentence needs reference
  2. Pg3. Study design and procedure: Sentence beginning with “The” instead of “Th”
  3. Pg5. Line 208: Authors should rectify typo, “meals” instead of “melas”.
  4. Authors should indicate significance in the all the figures wherever applicable for reader’s quick glance.
  5. Pg17. Line 467: Authors should rectify typo, “established” instead of “stablished”.

Author Response

Dear Reviewer,

Thank you for giving us the opportunity to submit a revised draft of our manuscript titled “Oat Polar Lipids Improve Cardiometabolic-related Markers After Breakfast and a Subsequent Standardized Lunch; A Randomized Crossover Study in Healthy Young Adults” to Nutrients (Manuscript ID: nutrients-1110462). We appreciate the time and effort that you have dedicated to providing your valuable feedback on our manuscript. We are grateful to you for your insightful comments on our paper. We have been able to incorporate changes to reflect most of the suggestions provided by you. We have highlighted the changes within the manuscript.

Here is a point-by-point response to your comments and concerns.

Overall Comment: In this age, where one of the biggest challenge and threat at the same time to humans is the ill-effects of unhealthy eating habits. Due to this, ultimately the life expectancy of people is on a constant decrease. Impetus is provided on eating healthy food products, superfoods, etc. As such this study becomes very crucial and it is my belief that it will aid us in recognizing healthy substitutes contributing to the our overall well being. The study shows beneficial effects of polar lipids from oat on postprandial glycemic regulation, blood lipids and gut hormones in healthy volunteers. This study involved evaluation of the effect of oat polar lipids on postprandial and second meal glycemic regulation, blood lipids gastrointestinal hormones, and subjective appetite-related variables in healthy humans. In the process, various parameters such as serum blood glucose concentrations, blood serum insulin concentrations, serum triglycerides, serum free fatty acid, ghrelin, GLP-1, etc were evaluated after breakfast and lunch. Authors strict control over various experimental parameters during the study design is commendable and appreciated. I read this prospective manuscript with great interest and accept it after addressing minor revisions as follows:

Response: We would like to thank you for such nice comments.

Minor specific comments:

Comment 1: Introduction line 42: End of sentence needs reference

Response: Revised as requested (p. 2, line 42).

Comment 2: Pg3. Study design and procedure: Sentence beginning with “The” instead of “Th”

Response: Done (p. 3, line 100).

Comment 3: Pg5. Line 208: Authors should rectify typo, “meals” instead of “melas”.

Response: Thank you. Corrected (p. 5 line 216).

Comment 4: Authors should indicate significance in the all the figures wherever applicable for reader’s quick glance.

Response: Thank you for this suggestion and we also basically agree on this. However, some of the graphs are very dense and indicating significance in the figures at different time points could result in difficulties to follow the actual lines due to much and overlapping letters.

Comment 5: Pg17. Line 467: Authors should rectify typo, “established” instead of “stablished”.

Response: Revised as requested (p. 17 line 476).

We look forward to hearing from you and to respond to any further questions and comments you may have.

Sincerely,

Mohammad Mukul Hossain

Lund University, Sweden.